# Evolved differences in larval social behavior mediated by novel pheromones

**Joshua D Mast[1]\*, Consuelo M De Moraes[2], Hans T Alborn[3], Luke D Lavis[1], David L Stern[1]**

[1]Janelia Research Campus, Howard Hughes Medical Institute, Ashburn, United States; [2]Department of Environmental Systems Science, ETH Zürich, Zürich, Switzerland; [3]Center for Medical, Agricultural, and Veterinary Entomology, USDA-ARS, Gainesville, United States

**Abstract** Pheromones, chemical signals that convey social information, mediate many insect social behaviors, including navigation and aggregation. Several studies have suggested that behavior during the immature larval stages of *Drosophila* development is influenced by pheromones, but none of these compounds or the pheromone-receptor neurons that sense them have been identified. Here we report a larval pheromone-signaling pathway. We found that larvae produce two novel long-chain fatty acids that are attractive to other larvae. We identified a single larval chemosensory neuron that detects these molecules. Two members of the *pickpocket* family of DEG/ENaC channel subunits (*ppk23* and *ppk29*) are required to respond to these pheromones. This pheromone system is evolving quickly, since the larval exudates of *D. simulans*, the sister species of *D. melanogaster*, are not attractive to other larvae. Our results define a new pheromone signaling system in *Drosophila* that shares characteristics with pheromone systems in a wide diversity of insects.

\*For correspondence: mastj@ janelia.hhmi.org

**Competing interests:** The authors declare that no competing interests exist.

**Reviewing editor**: Liqun Luo, Howard Hughes Medical Institute, Stanford University, United States

## Introduction

Different insect species use a diversity of pheromones to establish and organize their social encounters (*Karlson and Lüscher, 1959*; *Shorey, 1973*). For example, pheromones mediate courtship, aggression, alarm signaling, parental care, navigation, aggregation and many other behaviors (*Shorey, 1973*; *de Bruyne and Baker, 2008*). While the use of chemical cues is widespread amongst insects (*Symonds and Elgar, 2008*) and other taxa, including mammals (*Dulac and Torello, 2003*), the majority of our understanding of the molecular and neural mechanisms that mediate insect pheromone signaling has come from studies of a single species, *Drosophila melanogaster*. This is because many powerful genetic tools have been developed to study gene and neural circuit function in *Drosophila*. Most studies of pheromones in *Drosophila* have focused on pheromones related to sex (*Dahanukar and Ray, 2011*; *Fernández and Kravitz, 2013*). These studies have provided a rich mechanistic understanding of the receptor genes and sensory neurons that detect these cues, and of the circuit architectures that mediate these behaviors, linking the detection of specific molecules to appropriate behavioral outputs.

In contrast, pheromones that organize spatial behaviors like navigation or aggregation have been less intensively studied in *Drosophila* (*Bartelt et al., 1985*), despite the fact that such pheromones are critical for the survival of many insects. For example, ants employ substrate-born pheromones to establish and maintain foraging trails (*Steck, 2012*) and caterpillars use pheromones to organize mass migrations (*Fitzgerald, 1976*, *2003*). To date, it has not been possible to study such problems in *Drosophila* because no pheromones were known that exclusively mediate aggregation or trail following. Nonetheless, several previous studies have provided suggestive evidence that *Drosophila* larval behavior is influenced by social cues. First, *Drosophila* larvae are robustly attracted to odors produced by

**eLife digest** The release of chemical signals called pheromones is a common tactic used by animals in many social situations, such as to attract potential mates or to follow trails left by other members of their colony. Larvae of the fruit fly *Drosophila melanogaster*—a species commonly studied in the laboratory—gather together when sharing a food source and then cooperate in a way that may increase how efficiently they feed. It has been proposed that pheromones coordinate this behavior, but no larval pheromones had been identified.

Mast et al. noticed that Drosophila larvae crawling on a surface tended to occupy areas where other larvae had crawled before. This suggested that larvae had left attractive chemicals on the surface. Mast et al. identified two such substances by analyzing the chemicals left on the surface and then by testing the response of larvae to each compound.

Ultimately, Mast et al. found that a single sensory neuron in the larva is responsible for detecting these attractive chemical signals. Furthermore, two genes called *pickpocket23* and *pickpocket29* control this response. These genes were previously known for their roles in detecting sex pheromones, and they are members of a diverse family of calcium channel subunits that are involved in detecting multiple 'sensory modalities' such as touch and taste. When either *pickpocket23* or *pickpocket29* are inactivated, larvae ignore the social cues left by their neighbors.

Mast et al. also looked for an evolutionary role for these pheromones. Larvae of a closely related fly species called *Drosophila simulans* produce a subtly different blend of compounds to *D. melanogaster*, and this blend is not attractive to any of the species tested. While *Drosophila simulans* larvae were not attracted to the cues left by their own species, they were attracted to the pheromones produced by *Drosophila melanogaster*, indicating that they retain the sensory mechanisms to detect and respond to these pheromones. These results suggest that larvae experience a rapidly evolving, complex, pheromone-rich environment that may help them tailor their behavior to survive.

other larvae in food (*Durisko and Dukas, 2013*). Second, when in close proximity on a food source, larvae both aggregate (*Durisko et al., 2014*) and engage in a form of co-operative digging, which may effectively increase their rate of feeding (*Wu et al., 2003*; *Xu et al., 2008*). Finally, in natural conditions, where different species of drosophilids can be found sharing the same food resource, larvae preferentially pupariate with conspecific larvae and avoid pupariating with larvae of other species (*Beltramí et al., 2012*). Each of these three phenomena, it has been proposed, may be mediated by chemical cues produced by larvae or by larval activity.

We have discovered an attractive pheromone-signaling pathway in *Drosophila* larvae. We discovered two novel pheromones—*(Z)*-5-tetradecenoic acid and *(Z)*-7-tetradecenoic acid—that larvae deposit on substrate and that act as larval attractants. We identified a larval chemosensory neuron that is responsive to these compounds and that is required for behavioral attraction to these pheromones. We found that, in addition to their previously described roles in sex-pheromone detection, two members of the *pickpocket* family of *DEG/ENaC* channel subunits, *pickpocket23* and *pickpocket29*, are required for detecting *(Z)*-5-tetradecenoic acid and *(Z)*-7-tetradecenoic acid. Finally, we have found evolved differences in the production of these pheromones in the genus *Drosophila*, which, along with a repellent cue, cause differences in larval social behavior between these species. Our work provides the first window into social signaling during the larval stage of *Drosophila* and demonstrates that these mechanisms, and this social behavior, have evolved rapidly. Many insects employ fatty acid derived pheromones in the context of aggregation and navigation, and our work in *Drosophila* establishes a model system with which to further explore the sensory mechanisms and neural circuits that process this class of chemical cues.

## Results

### Drosophila larvae deposit an attractive cue on substrate

In preliminary assays of *Drosophila* larval behavior in dense populations, we noticed that larvae seemed to be attracted to regions of assay plates that had been occupied previously by other

larvae. We developed an assay to quantify this effect (*Figure 1A*). We allowed several hundred early third instar larvae to crawl on agarose plates for 20 min. Then, we cut away half the agarose surface, replaced it with fresh agarose, and placed a single new larva on the plate. We found that test larvae spent more time on the larval-treated substrate than on the untreated control substrate (*Figure 1B*), suggesting that the population of larvae had deposited an attractive pheromone on the substrate.

## A single receptor neuron necessary to respond to the attractive pheromone

We hypothesized that larvae detect the substrate-born cue through receptor neurons in the external larval chemosensory system. This system is composed primarily of the dorsal organ and the terminal organ, which are both located on the larval head and in total contain 67–72 sensory neurons (*Stocker, 2008*) (*Figure 1C*). We sought to address two questions. First, which sensory neurons in these organs, if any, mediate attraction to the larval-derived cue? Second, are any of the genes required for detecting adult sex pheromones required also to detect larval pheromones?

First, we tested if the substrate-born attractant is detected by larval neurons expressing specific odorant receptor or gustatory receptor genes that mediate adult social behaviors like courtship and aggression (*Kurtovic et al., 2007*; *Miyamoto and Amrein, 2008*; *Moon et al., 2009*; *Wang and Anderson, 2010*; *Liu et al., 2011*; *Wang et al., 2011*). Silencing synaptic transmission by expressing tetanus toxin (*Sweeney et al., 1995*) in *Orco-GAL4, Gr32a-GAL4,* or *Gr33a-GAL4* expressing cells failed to block attraction to the larval cue (*Figure 1—figure supplement 1*). Similarly, mutations in these receptor genes had no effect on this behavior (*Figure 1—figure supplement 1*). Thus, larval neurons expressing odorant receptors and the gustatory receptors *Gr32a* and *Gr33a* are unlikely to be required to mediate attraction to the larval pheromone.

Several members of the *pickpocket* family of *degenerin/DEG/ENaC* channel subunit genes are expressed in gustatory neurons on the leg segments and labellum of adult flies and are required for detecting sex pheromones (*Liu et al., 2012*; *Lu et al., 2012*; *Thistle et al., 2012*; *Toda et al., 2012*; *Yuan et al., 2013*). We tested if two members of this family, *pickpocket23 (ppk23)* and *pickpocket29 (ppk29)*, are required also for the detection of the attractive larval cue. First, we tested *ppk23*. Silencing *ppk23-GAL4* expressing neurons with tetanus toxin reduced the attraction of larvae to the cue, as did a mutation deleting the *ppk23* locus, Δ*ppk23* (*Figure 1B*). *ppk23-GAL4* is expressed in 10 neurons that innervate the terminal organ, the main larval taste organ (*Figure 1C,D*), and in one internal sensory neuron located along the larval pharynx (*Figure 1—figure supplement 2A*). A subset of sensory neurons that innervate the terminal organ have their cell bodies located in the dorsal organ ganglion (TO dorsolateral neurons), while the remaining terminal organ cells are housed in the terminal organ ganglion (*Colomb et al., 2007*). *ppk23* labels all three TO dorsolateral neurons. *ppk23* neurons project widely throughout the subesophageal zone (*Figure 1E*), the brain region that integrates sensory information related to taste and feeding (*Vosshall and Stocker, 2007*; *Stocker, 2008*).

Next, we examined *ppk29*. Larvae carrying a deletion of the *ppk29* gene displayed reduced attraction to larval residue compared with controls (*Figure 1B*). We examined *ppk29-GAL4* expression and found that it is expressed ubiquitously in larvae (*Figure 1—figure supplement 2B,C*) and therefore is not useful for defining the requirement for *ppk29* specific sensory neurons. Together, these results indicate that both *ppk23* and *ppk29* are required for attraction of larvae to larval residue.

To refine the identity of candidate pheromone receptor neurons further, we screened a subset of the Janelia enhancer fragment *GAL4* lines (*Jenett et al., 2012*) that were characterized as driving sparse patterns of expression in anterior sensory neurons (*Li et al., 2013*). We drove tetanus toxin with each line and found that neuronal inactivation in several lines blocked the larval attraction to larval residue (*Figure 1B* and *Figure 1—figure supplement 3*). We focused further experiments on one of these lines that failed to respond to the larval residue, *R58F10,* and that drives expression in only a single bilateral TO dorsolateral neuron (*Figure 1C,F*). This neuron is therefore a subset of the three TO dorsolateral neurons marked by *ppk23*. *R58F10* neurons send straight, mostly unbranched projections into the subesophagael zone (*Figure 1G*).

## Identification of the attractive larval pheromone

No molecules that mediate social interactions between *Drosophila* larvae are known. To identify the molecules that larvae may use as attractive pheromones, we took advantage of the fact that the attractive activity was deposited on substrate by crawling larvae. We found that the activity could be extracted

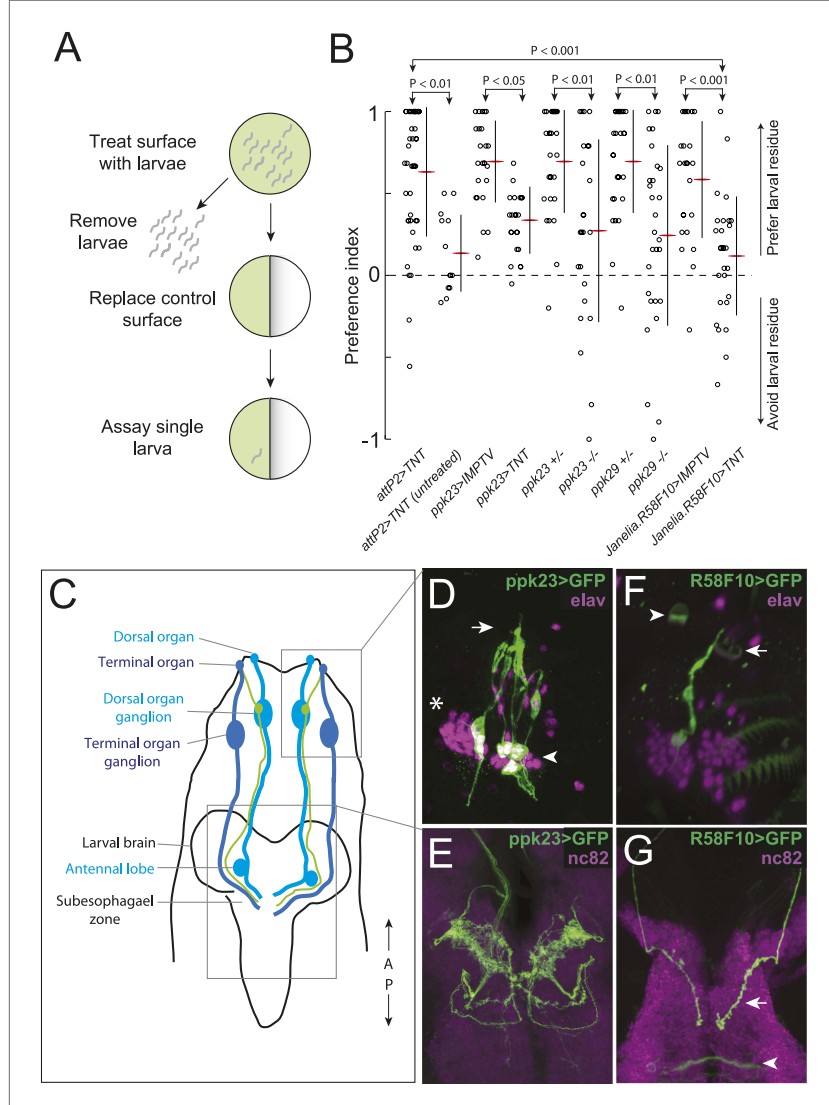

**Figure 1**. A single class of chemosensory receptors is required for larvae to respond to attractive larval-derived cues. (**A**) An assay to quantify larval attraction to substrate-born attractive pheromone. A high density of wild-type early third instar larvae were allowed to crawl over the surface of an agarose plate for 20 min. The larvae and half of the treated agarose (green) were removed and replaced with fresh control agarose (white). The fraction of time spent on each surface by single larvae was used to calculate a preference index score (Preference Index = (time on test substrate—time on control substrate)/total time). (**B**) A behavioral neuronal-silencing screen to identify the receptor neurons required for response to larval attractive pheromones. Data points represent the preference index scores of individual larvae in response to the larval cue in this assay. In this and subsequent figures, horizontal small lines represent means and vertical lines represent ± one standard deviation. Genotypes displayed significant heterogeneity (ANOVA, $F_{9,255}$ = 9.75, p < 0.0001). Control larvae carrying the empty *attP2* landing site with no *GAL4* insertion and a copy of *UAS-tetanus toxin (UAS-TNT)* were attracted to the larval cue, but not to untreated substrate. Silencing neurons expressing *ppk23* with *TNT* blocked the response to the cue. Driving an inactive variant of tetanus toxin, *IMPTV,* with *ppk23-GAL4* did not block attraction. Mutations affecting *ppk23* or *ppk29* also blocked attraction. Silencing neurons with Janelia enhancer fragment *R58F10-GAL4* blocked the response to the cue. Driving *IMPTV* with *R58F10-GAL4* did not block attraction. In this and subsequent figures, statistical comparisons between treatments were performed using the Tukey honest significant difference test and are reported for comparisons significant at p < 0.05 or below. (**C**) A schematic of neurons comprising the main external sensory organs in the larval chemosensory system. Receptor neurons in the terminal organ ganglion (TOG; dark blue) send sensory terminals into the terminal organ (TO), and synaptic terminals into the subesophageal zone (SEZ). Receptor neurons in the dorsal organ ganglion (DOG; light blue) send sensory terminals mainly into the dorsal organ (DO), and synaptic

*Figure 1. Continued on next page*

*Figure 1. Continued*

terminals into both the antennal lobe and the SEZ. Three neurons (green) in the DOG send sensory terminals into the dorsolateral papillum of the TO, and project to the SEZ. (**D**) *ppk23* expressing neurons innervating the terminal organ (arrow), marked by membrane-targeted green fluorescent protein, mCD8:GFP (green, anti-GFP). Neuron cell bodies in the dorsal organ ganglion and terminal organ ganglion are labeled with anti-elaV (magenta). The TOG and DOG are marked by an arrowhead and asterisk, respectively. (**E**) The axon terminals of *ppk23* expressing neurons in the larval brain (green, anti-GFP). Neuropil is marked by antibodies against the perisynaptic marker *bruchpilot* (nc82, magenta). (**F**) *R58F10* sensory neurons innervate the terminal organ (arrow). R58F10 marked by membrane-targeted green fluorescent protein, mCD8:GFP (anti-GFP, green; elaV, magenta). The dorsal organ is indicated with an arrowhead. (**G**) *R58F10* neuron synaptic terminals project into the suboesophageal zone (arrow). Secondary lineage neurons in the ventral nerve cord are also marked by *R58F10* (arrowhead) (anti-GFP, green; nc82, magenta).

The following figure supplements are available for figure 1:

**Figure supplement 1**. Sex-pheromone receptor genes with known larval expression are not required to respond to attractive larval pheromones.

**Figure supplement 2**. The expression patterns of *ppk23-GAL4* and *ppk29-GAL4* in larval sensory organs and the central nervous system.

**Figure supplement 3**. A synaptic silencing screen to identify sensory neurons required for attractive pheromone-driven behavior.

off glass upon which larvae had crawled using either hexane or acetone as solvent (*Figure 2A,B*). Gas chromatography-mass spectrometry of these extracts revealed that they contained seven common saturated fatty acids and fatty acid derivatives and two rare fatty acid monoenes, (*Z*)-5-tetradecenoic acid and (*Z*)-7-tetradecenoic acid (*Figure 2C*, *Figure 5—source data 1*). We did not detect the adult sex pheromones (*Z*)-11-vaccenyl acetate (cVA), 7-tricosene, 7,11-heptocosadiene, or CH503 in these extracts. While we observed (*Z*)-9-tetradecenoic acid in extracts of whole larval bodies, we did not observe significant levels of this compound in the residue deposited by larvae, suggesting that (*Z*)-5-tetradecenoic acid and (*Z*)-7-tetradecenoic acid are the major tetradecanoic derivatives deposited by larvae. (*Z*)-5-tetradecenoic acid is found also as a component of a defensive, ant-repulsive secretion released by some thrips (*Suzuki et al., 2004*). To our knowledge, (*Z*)-7-tetradecenoic acid has not been found in any other insect.

To determine which, if any, of the nine compounds deposited by larvae acts as a larval attractant, we measured the attractive activity of synthetic versions of each molecule using a checkerboard-layout behavioral assay (*Figure 2D*), which allowed us to assay larvae at a density similar to the single plate assay (*Figure 1A*), but at a higher throughput. Control larvae were attracted to low concentrations (50 fm/cm$^2$) of (*Z*)-5-tetradecenoic acid and (*Z*)-7-tetradecenoic acid, but not to any other compound (*Figure 2E* and *Figure 2—figure supplement 1*). Moreover, larvae showed a sustained response to these two molecules over the entire duration of a 5-min assay (*Figure 2F,G*). Dose response curves revealed that (*Z*)-5-tetradecenoic acid and (*Z*)-7-tetradecenoic acid are strongly attractive to larvae across a broad range of concentrations (*Figure 2H*). Larvae responded most strongly to pheromone concentrations of 50 fm/cm$^2$ and this effect decreased at higher concentrations. A similar phenomenon has been observed for pheromones in multiple other species where higher doses induce a less robust behavioral response, or even begin to elicit alarm or escape behaviors, compared with lower doses (*Leal et al., 1989*; *Shimizu et al., 2003*). Finally, we assayed the larval response to an equimolar mixture ((25 fmol:25 fmol)/cm$^2$) of the two compounds. Larvae were attracted to a blend of these compounds, but were less responsive to this mixture than to a presentation of each molecule alone, which suggests an interaction between the two pheromones (*Figure 2I,J*). These results indicate that larvae deposit the two compounds (*Z*)-5-tetradecenoic acid and (*Z*)-7-tetradecenoic acid on substrate and that low concentrations of these compounds provide an attractive cue to other larvae.

### *R58F10* neurons detect larval attractive pheromones

Many of the nine compounds identified in larval residue when tested alone at high concentrations (0.5 nmol/cm$^2$) were at least mildly attractive to larvae (*Figure 2—figure supplement 3*). This

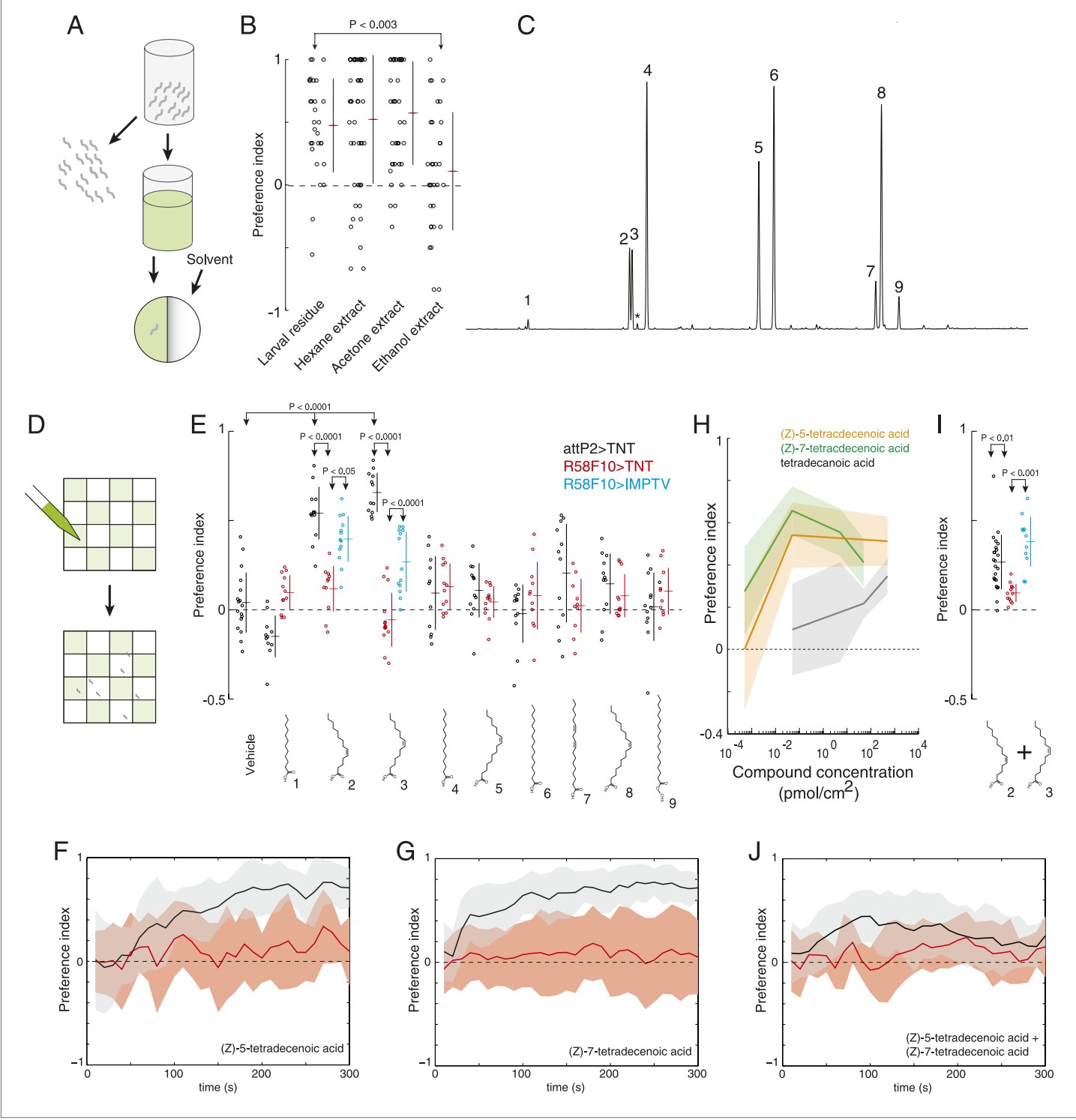

**Figure 2**. (*Z*)-5-tetradecenoic acid and (*Z*)-7-tetradecenoic acid are attractive larval pheromones. (**A**) A method for extracting the attractive cue deposited on substrate. Larvae were allowed to crawl over the inner surface of a glass vial for 30 min. Larvae were removed and replaced by solvent. Treated solvent was deposited on one side of an agarose plate, and untreated solvent on the other. (**B**) The attraction of wild-type *D. melanogaster* larvae to extracts from larval-treated glass using hexane, acetone, or ethanol as solvent compared to control solvent. Data points are the preference index scores of individual wild-type larvae. Hexane and acetone extracts elicited similar levels of attraction as plates treated directly with larvae. Ethanol extracts did not. (**C**) Chromatograph from GC analysis of *D. melanogaster* larval hexane extracts. The major components are saturated fatty acids, fatty acid monoenes and dienes. Each peak is labeled as follows: (1) dodecanoic acid, (2) (*Z*)-5-tetradecenoic acid, (3) (*Z*)-7-tetradecenoic acid, (4) tetradecanoic acid,

*Figure 2. Continued on next page*

*Figure 2. Continued*

(5) (*Z*)-9-hexadecenoic acid, (6) hexadecanoic acid, (7) (*Z,Z*)-9,12-octadecadienoic acid, (8) (*Z*)-9-octadecenoic acid, and (9) octadecanoic acid, methyl ester. Small amounts of (*Z*)-9-tetradecenoic acid (*) were also detected between peaks (3) and (4). (**D**) Schematic of a multiple-larva assay to measure the attractive activity of individual compounds identified from larval residue. Test surfaces were coated in a checkerboard pattern with candidate molecules (green) or vehicle solvent (white). (**E**) A behavioral screen measuring the attraction of control (*attP2;UAS-TNT*, black), *R58F10-GAL4;UAS-TNT* (red), and *R58F10-GAL4; UAS-IMPTV* (blue) larvae to individual compounds found in the *D. melanogaster* larval extracts. Data points are the preference index scores of small groups (10-15 individuals) of larvae in response to 50 fmol/cm$^2$ of candidate molecule. Compound treatments (ANOVA; $F_{9,113}$ = 22.3, p < 0.0001) and genotype by compound interactions (2-way ANOVA; $F_{1,232}$ = 17.7, p < 0.0001) displayed significant heterogeneity. Larvae were attracted to (*Z*)-5-tetradecenoic acid and (*Z*)-7-tetradecenoic acid. Silencing the synaptic activity of the sensory neurons expressing *R58F10-GAL4* strongly inhibited the response to these molecules. Attraction was partially restored when driving the inactive *TNT* gene product, *IMPTV*. (**F**) The response of *attP2;UAS-TNT* (black), and *R58F10-GAL4;UAS-TNT* (red) larvae to 50 fmol/cm$^2$ (*Z*)-5-tetradecenoic acid over time. (**G**) The timecourse of response to 50 fmol/cm$^2$ (*Z*)-7-tetradecenoic acid. Solid lines represent the mean preference index score at each timepoint. Shaded areas represent ± one standard deviation. (**H**) Dose response curves of larval preferences to (*Z*)-5-tetradecenoic acid (orange), (*Z*)-7-tetradecenoic acid (green), and saturated tetradecanoic acid (grey). Shaded areas represent ± one standard deviation. (**I**, **J**) The preference index scores (**I**) and preference time course (**J**) of *attP2;UAS-TNT* (black), *R58F10-GAL4;UAS-TNT* (red), and *R58F10-GAL4; UAS-IMPTV* (blue) larvae in response to an equimolar mixture ((25 fmol:25 fmol)/cm$^2$) of (*Z*)-5-tetradecenoic and (*Z*)-7-tetradecenoic acids.

The following figure supplements are available for figure 2:

**Figure supplement 1**. Time courses of the attraction of larvae to individual compounds identified from larval extracts at 50 fmol/cm$^2$.

**Figure supplement 2**. Chemical analysis of synthetic and natural pheromones.

**Figure supplement 3**. A behavioral screen measuring the attraction of larvae to individual compounds at 0.5 nmol/cm$^2$.

concentration corresponds to the concentration of compounds in extract of larval residue generated by hundreds of individuals. Thus, in principle, many or all of these larval compounds might contribute to larval social behavior. To determine which, if any, of these compounds are detected by *R58F10* neurons, we tested the behavioral response of larvae to synthetic compounds with and without silenced *R58F10* neurons. Silencing *R58F10* neurons with tetanus toxin blocked larval attraction to (*Z*)-5-tetradecenoic acid, (*Z*)-7-tetradecenoic acid, and a mixture of the two pheromones, but had no significant effect on the response to the other compounds at 50 fmol/cm$^2$ (*Figure 2E,I*). Driving a mutant form of tetanus toxin carrying two point mutations that block its cleavage of synaptobrevin (*IMPTV*) caused an intermediate response to these pheromones. This suggests that some of the effect of driving *TNT* in *R58F10* may arise from indirect toxicity. Other groups have also found that expressing IMPTV can sometimes produce intermediate phenotypes (*Kain et al., 2013*). These results demonstrate that larval attraction to previously occupied substrate can be mediated by the sensitivity of *R58F10* neurons to (*Z*)-5-tetradecenoic acid and (*Z*)-7-tetradecenoic acid.

We next assayed whether *R58F10* sensory neurons respond directly to (*Z*)-5-tetradecenoic acid and (*Z*)-7-tetradecenoic using a genetically encoded activity reporter, GCaMP6 (*Chen et al., 2013*). We tested a range of pheromone concentrations estimated to encompass those encountered by larvae in the previously described behavioral assays. *R58F10* neurons were excited in response to (*Z*)-5-tetradecenoic acid at 0.01 nM, 1 nM, and 100 nM. *R58F10* neurons were not significantly excited by (*Z*)-7-tetradecenoic acid at 0.01 nM, but responded to higher concentrations (*Figure 3A,B*). Moreover, at 1 nM, which corresponds to the concentration that evoked the peak behavioral preference (*Figure 2H*), *R58F10* neurons exposed to (*Z*)-5-tetradecenoic acid or (*Z*)-7-tetradecenoic acid were strongly elevated in their integrated fluorescence over the length of the trial compared to vehicle controls (*Figure 3C*). We did not observe a significant increase in the integrated fluorescence when the compounds were tested at 0.01 nM and 100 nM (*Figure 3C*), suggesting that the dose-response relationship of *R58F10* to these pheromones is complicated.

We explored the specificity of *R58F10* neurons for (*Z*)-5-tetradecenoic acid and (*Z*)-7-tetradecenoic acid by testing if other compounds with similar structure that were present in the larval extracts could evoke activity in these cells. *R58F10* neurons were not excited by tetradecanoic acid, which lacks a double bond compared to these pheromones, by (*Z*)-9-octadecanoic acid, a structurally similar, but longer, fatty acid monoene, nor by vehicle controls (*Figure 3A–C*). These data demonstrate that

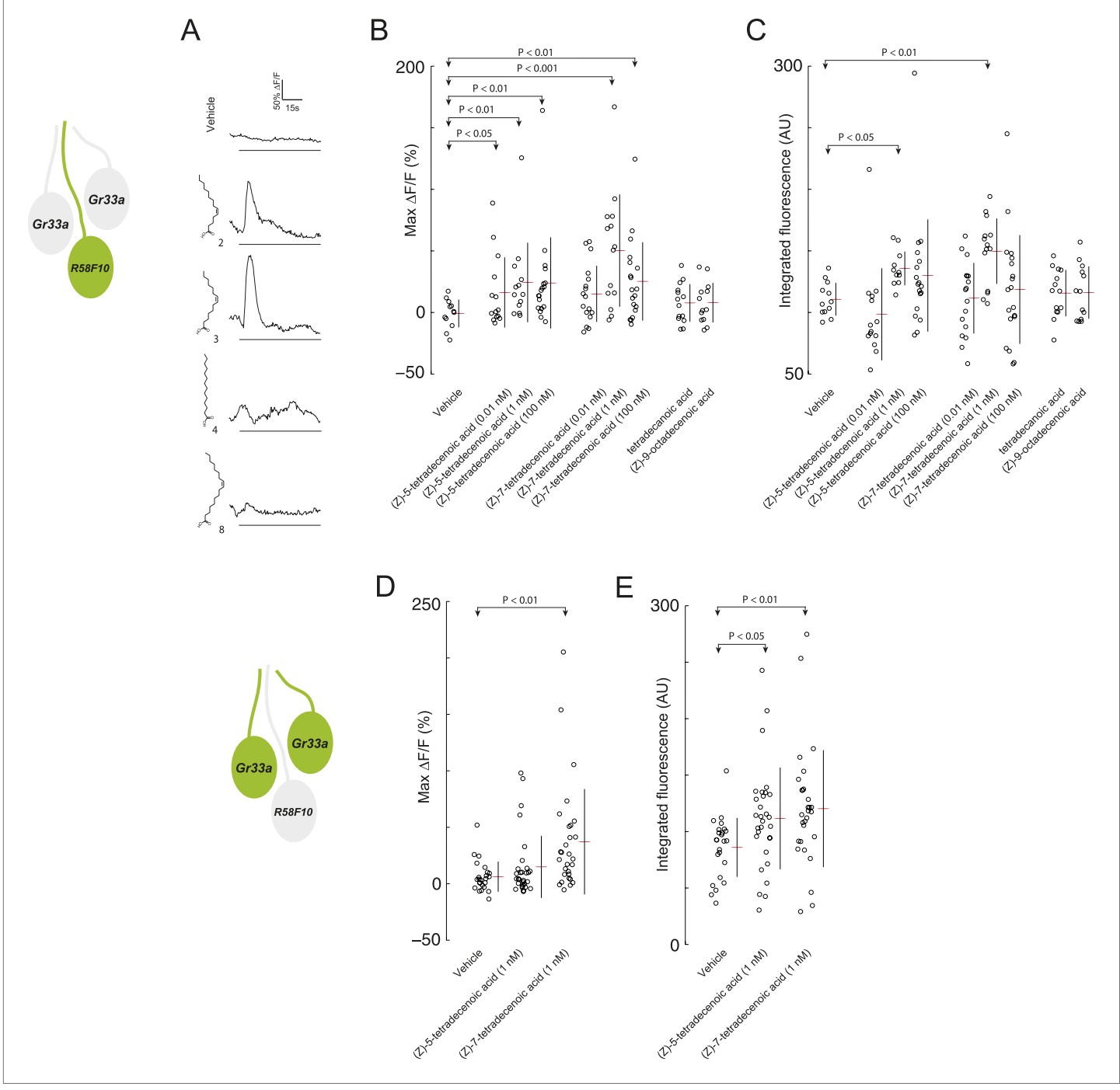

**Figure 3**. *R58F10* sensory neurons are excited by (*Z*)-5-tetradecenoic acid and (*Z*)-7-tetradecenoic acid. (**A**) Example traces of the excitation of *R58F10* sensory neurons in response to control vehicle, 1 nM (*Z*)-5-tetradecenoic acid, (*Z*)-7-tetradecenoic acid, tetradecanoic acid, or (*Z*)-9-octadecanoic acid measured using the calcium–sensitive activity reporter, GCaMP6-S, over time. Excitation is reported as the percent change in GCaMP fluorescence over baseline (%ΔF/F). The solid black bar along the x-axis signifies the onset and duration of stimulus. (**B** and **C**) Stimulation with either (*Z*)-5-tetradecenoic acid or (*Z*)-7-tetradecenoic acid across a broad range of concentrations evoked a significant increase in the maximum percent change in GCaMP fluorescence (max %ΔF/F) (**B**) and in the integrated fluorescence (arbitrary units) (**C**) of individual trials over 150 s compared to control buffer. Compound treatments displayed significant heterogeneity (ANOVA; $F_{8,125}$ = 3.26, p < 0.01; ANOVA; $F_{8,123}$ = 3.02, p < 0.01). Data points represent recordings from individual cells. (**D** and **E**) The stimulation of two other *ppk23*-expressing sensory neurons innervating the dorsolateral terminal organ sensillum with *R58F10*, marked by *Gr33a-GAL4* with 1 nM (*Z*)-5-tetradecenoic acid or (*Z*)-7-tetradecenoic acid evoked a significant increase in the maximum percent change in GCaMP fluorescence (max %ΔF/F) (**D**) and in the integrated fluorescence (arbitrary units) (**E**), compared to control. Compound treatments displayed significant heterogeneity (ANOVA; $F_{2,80}$ = 6.15, p < 0.01; ANOVA; $F_{2,80}$ = 4.23, p < 0.05). Data points represent recordings from individual cells. Data are pooled for both cells.

*R58F10* sensory neurons are directly excited by, and exhibit a high degree of specificity for, the two larval pheromones (*Z*)-5-tetradecenoic acid and (*Z*)-7-tetradecenoic acid.

We wondered whether (*Z*)-5-tetradecenoic acid and (*Z*)-7-tetradecenoic acid are detected only by *R58F10*, or if other sensory neurons—for example, other neurons that express *ppk23*—are also responsive to these compounds. Specifically, we tested if the two neurons that innervate the dorsolateral terminal organ sensillum along with *R58F10* are sensitive to larval pheromone using GCaMP6. Both of these cells are marked by *Gr33a-GAL4* expression, and as noted previously, also express *ppk23*. We were unable to morphologically discriminate between these two cells, so we pooled the individual fluorescence data from both cells. Both (*Z*)-5-tetradecenoic acid and (*Z*)-7-tetradecenoic acid elicited activity in at least one of these neurons (*Figure 3D,E*). While neither cell is necessary for the attraction to larval residue (*Figure 1—figure supplement 1*), these cells may control other aspects of larval behavior driven by these compounds.

## An expanded role for *pickpocket23* and *pickpocketk29* in larval pheromone detection

Since we observed that both *ppk23* and *ppk29* were required for larvae to respond to a full bouquet of larval residue (*Figure 1*), we next tested if these genes were required for detecting (*Z*)-5-tetradecenoic and (*Z*)-7-tetradecenoic acid specifically. The *ppk23-GAL4* expression pattern includes the larval pheromone sensing neuron *R58F10*. As expected, silencing *ppk23-GAL4* expressing neurons with tetanus toxin reduced larval attraction to both (*Z*)-5-tetradecenoic acid (*Figure 4A,B*) and (*Z*)-7-tetradecenoic acid (*Figure 4E,F*) compared to control larvae. Removing *ppk23* gene function also reduced the attraction of larvae to both of these compounds compared to heterozygous controls (*Figure 4A,C,E,G*). We next tested if *ppk23* function is directly required in the sensory neuron *R58F10* by rescuing *ppk23* expression specifically with *R58F10-GAL4* in a *Δppk23* mutant background. The expression of *ppk23* in *R58F10* cells partially restored the attraction of larvae to both larval pheromones (*Figure 4A,D,E,H*). These results indicate that *ppk23* gene function is required specifically in *R58F10* neurons for response to the larval pheromones (*Z*)-5-tetradecenoic acid and (*Z*)-7-tetradecenoic acid.

We tested next if *ppk29* is required for detecting these larval pheromones. Larvae homozygous for a deletion of *ppk29* were not attracted to (*Z*)-5-tetradecenoic acid (*Figure 4I,J*) and displayed weaker attraction to (*Z*)-7-tetradecenoic acid (*Figure 4L,M*) than did heterozygous control larvae. The expression of *ppk29* in *R58F10* cells did not rescue the attraction of larvae to (*Z*)-5-tetradecenoic acid (*Figure 4I,K*), but partially restored the attraction of larvae to (*Z*)-7-tetradecenoic acid (*Figure 4L,N*). These results indicate that *ppk29* gene function is required specifically in *R58F10* neurons for response to the larvae pheromone (*Z*)-7-tetradecenoic acid. That *ppk29* expression in *R58F10* failed to rescue larval attraction to (*Z*)-5-tetradecenoic acid indicates that either *ppk29* is required in another cell, or that the expression of *ppk29* by *R58F10*-GAL4 is inadequate to rescue this behavior.

Finally, we performed calcium imaging to test if *ppk23* and *ppk29* are required for *R58F10* sensory neurons to be excited by these pheromones. *R58F10* neurons mutant for either *ppk23* or *ppk29* were not excited by 1 nM (*Z*)-5-tetradecenoic acid nor by (*Z*)-7-tetradecenoic acid (*Figure 4O,P*). Due to technical limitations we were unable to identify *ppk23* heterozygotes that would serve as an appropriate positive control for this experiment. All together, these results demonstrate that both *ppk23* and *ppk29* are required for robust attraction to (*Z*)-5-tetradecenoic acid and (*Z*)-7-tetradecenoic acid. To our knowledge, this is the first identification of genes required for pheromone detection at the larval stage in any insect.

## Evolved differences in larval pheromone signaling

Pheromone signals often evolve quickly between species (*Symonds and Elgar, 2008*), including between species of the genus *Drosophila* (*Jallon and David, 1987*; *Ferveur, 2005*; *Shirangi et al., 2009*). To determine if larval pheromone signaling has evolved within the *D. melanogaster* species group, we investigated social behavior in two sister species of *D. melanogaster*, *Drosophila simulans* and *Drosophila sechellia*. These species diverged from *D. melanogaster* approximately 3–5 MYA and diverged from each other approximately 0.5 MYA (*Kliman et al., 2000*; *Lachaise and Silvain, 2004*). First, we tested if *D. sechellia* and *D. simulans* larvae deposit residue that is attractive to other larvae. We found that third instar *D. sechellia* larvae are attracted, and *D. simulans* larvae are not attracted, to substrate treated with conspecific larvae (*Figure 5*). These data are consistent with recent independent observations that *D. simulans* larvae are less likely to form aggregations than are *D. melanogaster* larvae (*Durisko et al., 2014*). We reasoned that this difference could be caused by evolved changes

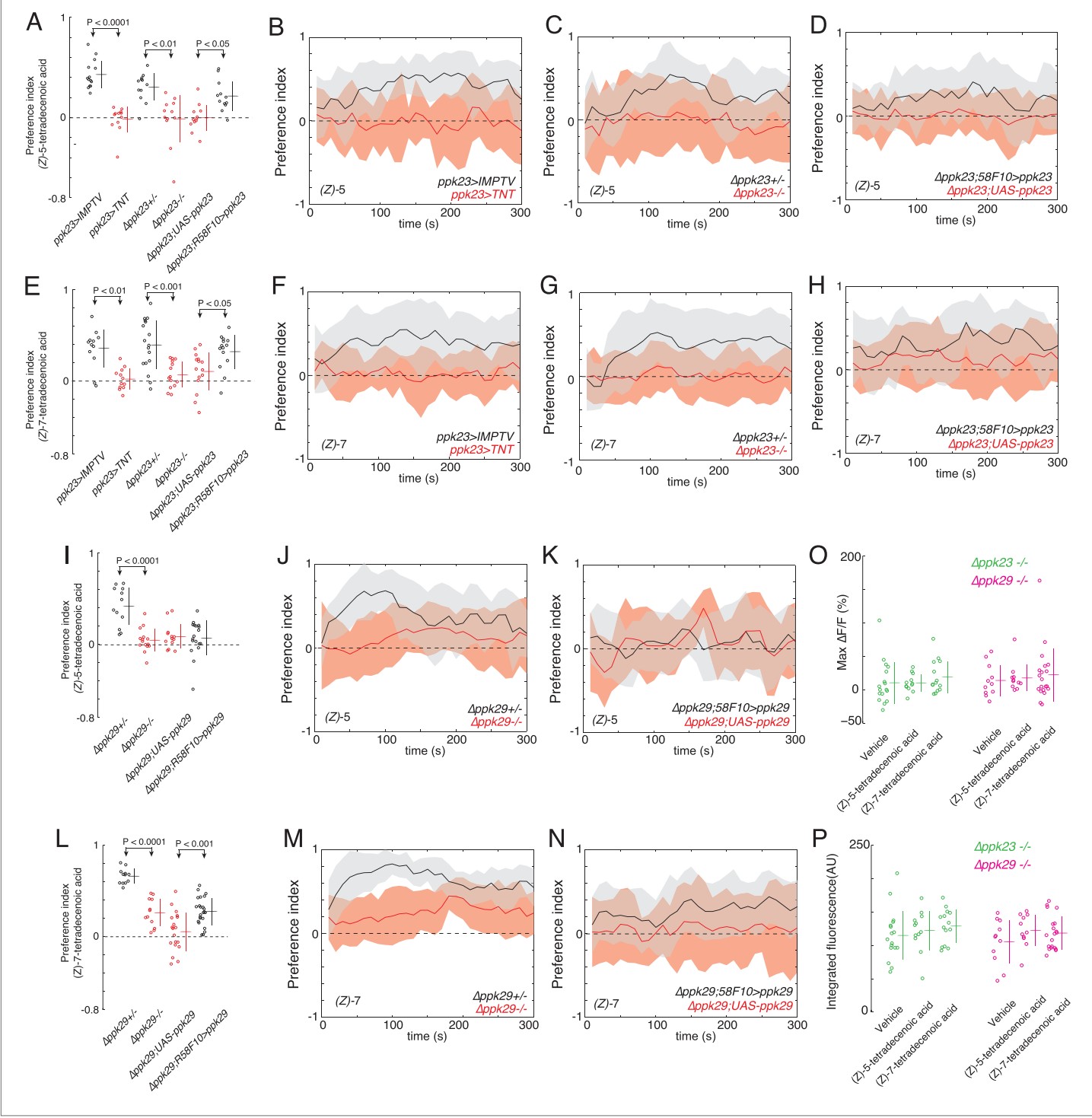

**Figure 4**. *ppk23* and *ppk29* are required for detecting larval pheromones. (**A** and **E**) The mean index scores from preference assays measuring the attraction of control, *ppk23* silenced, *Δppk23* mutant, and *Δppk23* mutant larvae rescued with *ppk23* expression specifically in *R58F10* neurons to synthetic (Z)-5-tetradecenoic acid (**A**) or (Z)-7-tetradecenoic acid (**E**). Data points are the mean preference index scores of small groups (10-15 individuals) of larvae in response to 50 fmol/cm² of candidate molecule over the course of the 300 s assay. Genotypes displayed significant heterogeneity (ANOVA; $F_{5,68} = 18.03$, $p < 0.0001$, $F_{5,78} = 9.03$, $p < 0.0001$, respectively). The mean preference index over the time-course of the larval response to either (Z)-5-tetradecenoic acid (**B**–**D**) or (Z)-7-tetradecenoic acid (**F**–**H**). Silencing the synaptic activity of *ppk23* expressing neurons, as well as a mutation deleting *ppk23* blocked the attraction of larvae to (Z)-5-tetradecenoic acid and (Z)-7-tetradecenoic acid. The mean preference index scores

*Figure 4. Continued on next page*

*Figure 4. Continued*

(**I**, **L**) and timecourse (**J**, **K**, **M**, **N**) from preference assays measuring the attraction of control and Δ*ppk29* mutant larvae to synthetic (*Z*)-5-tetradecenoic acid (**I**, **J**, **K**) or (*Z*)-7-tetradecenoic acid (**L**, **M**, **N**). A mutation deleting *ppk29* blocked the attraction of larvae to both (*Z*)-5-tetradecenoic acid (p < 0.001) and (*Z*)-7-tetradecenoic acid (p < 0.001). Stimulation of *R58F10* neurons mutant for *ppk23* or *ppk29* with either (*Z*)-5-tetradecenoic acid or (*Z*)-7-tetradecenoic acid evoked no significant increase in the maximum percent change in GCaMP fluorescence (max %ΔF/F) (ANOVA; $F_{5,79}$ = 0.45, p < 0.83) (**O**) or in the integrated fluorescence (arbitrary units) (ANOVA; $F_{5,79}$ = 0.90, p < 0.81) (**P**) of individual trials over 150 s compared to control buffer.

in *D. simulans* leading to either (1) loss of sensitivity to an attractant, (2) loss of production of an attractant, (3) production of a repellent molecule, or a combination of these factors.

To determine if *D. simulans* larvae have lost sensitivity to an attractive pheromone, we tested whether *D. simulans* larvae were attracted to residue produced by *D. melanogaster* and *D. sechellia*. *D. simulans* larvae were as attracted to these residues, as were *D. melanogaster* and *D. sechellia* (*Figure 5A*). Furthermore, *D. simulans* larvae were attracted to synthetic (*Z*)-5-tetradecenoic acid and (*Z*)-7-tetradecenoic acid, and their preference was indistinguishable from the attraction of *D. melanogaster* and *D. sechellia* to these molecules (*Figure 2E*, *Figure 5A*; *Figure 5—figure supplement 1*). Together, these data demonstrate that sensitivity to both (*Z*)-5-tetradecenoic acid and (*Z*)-7-tetradecenoic acid is unimpaired in *D. simulans.*

Next, to determine if *D. simulans* does not produce attractive pheromones, we tested first if the other species were attracted to *D. simulans* residue. We found that both *D. melanogaster* and *D. sechellia* larvae were indifferent to *D. simulans* residue, but were attracted to *D. sechellia* residue (*Figure 5A*). To determine if (*Z*)-5-tetradecenoic acid and (*Z*)-7-tetradecenoic acid production was specifically lost or attenuated in *D. simulans*, we tested if an attractive cue could be extracted from glass treated with larvae of these species using either hexane or acetone as a solvent. Extracts from glass treated with *D. simulans* larvae were not attractive to wild-type *D. melanogaster* larvae. In contrast, both *D. sechellia* hexane and acetone extracts were attractive to wild-type *D. melanogaster* larvae (*Figure 5A*). We found that levels of both (*Z*)-5-tetradecenoic acid and (*Z*)-7-tetradecenoic acid are reduced in *D. simulans* relative to *D. sechellia* and *D. melanogaster* (*Figure 5B,C*, *Figure 5—source data 1*). These results suggest that larvae of *D. simulans* produce less of the attractive cues (*Z*)-5-tetradecenoic acid and (*Z*)-7-tetradecenoic acid than do *D. melanogaster* and *D. sechellia* larvae.

Finally, we explored whether *D. simulans* produces a repellent cue. This hypothesis was supported, initially, by the observation that acetone extract of *D. simulans* larval residue was mildly repulsive to *D. simulans* larvae (*Figure 6B,C*). We reasoned that if *D. simulans* produced a repellent, then this repellent may be able to block the attractiveness of (*Z*)-5-tetradecenoic acid and/or (*Z*)-7-tetradecenoic acid. We tested this hypothesis by supplementing *D. simulans* larval extract with the attractive pheromones (*Figure 6A*). While (*Z*)-5-tetradecenoic acid alone was attractive to *D. simulans*, this activity was blocked completely by the addition of *D. simulans* extract (*Figure 6B,D*). These data are consistent with the hypothesis that an unidentified repellent is present in the *D. simulans* extract. In contrast, the addition of (*Z*)-7-tetradecenoic acid to *D. simulans* extract partially rescued the attraction of *D. simulans* to the extract, but not to the level observed for (*Z*)-7-tetradecenoic acid tested alone (*Figure 6B,E*). Last, while *D. simulans* larvae were attracted to a mixture of the two pheromones alone, the addition of both pheromones to *D. simulans* extract conferred no attractive activity (*Figure 6B,F*). Taken together, these results suggest that the reduced levels of (*Z*)-7-tetradecenoic acid production, combined with an unidentified repellent cue released by *D. simulans,* generate the evolved differences in larval social behavior between these closely related species. Furthermore, the unidentified repellent cue may interact with (*Z*)-5-tetradecenoic acid to overcome the residual attractive activity of (*Z*)-7-tetradecenoic acid.

## Discussion

We have demonstrated that (*Z*)-5-tetradecenoic acid and (*Z*)-7-tetradecenoic acid are attractive or possibly arrestant larval pheromones in *D. melanogaster* that are detected by a sensory neuron marked by the enhancer fragment *R58F10*. This is the first direct evidence of pheromone-driven behavior in *Drosophila* larvae, and this system promises to serve as a model for dissecting the role of pheromones in navigation and aggregation. How *Drosophila* larvae use (*Z*)-5-tetradecenoic acid and (*Z*)-7-tetradecenoic acid in a natural setting remains an open question. These molecules could be released locally to enhance larval aggregation at a food source, since *Drosophila* larvae, like many other insect species, exploit

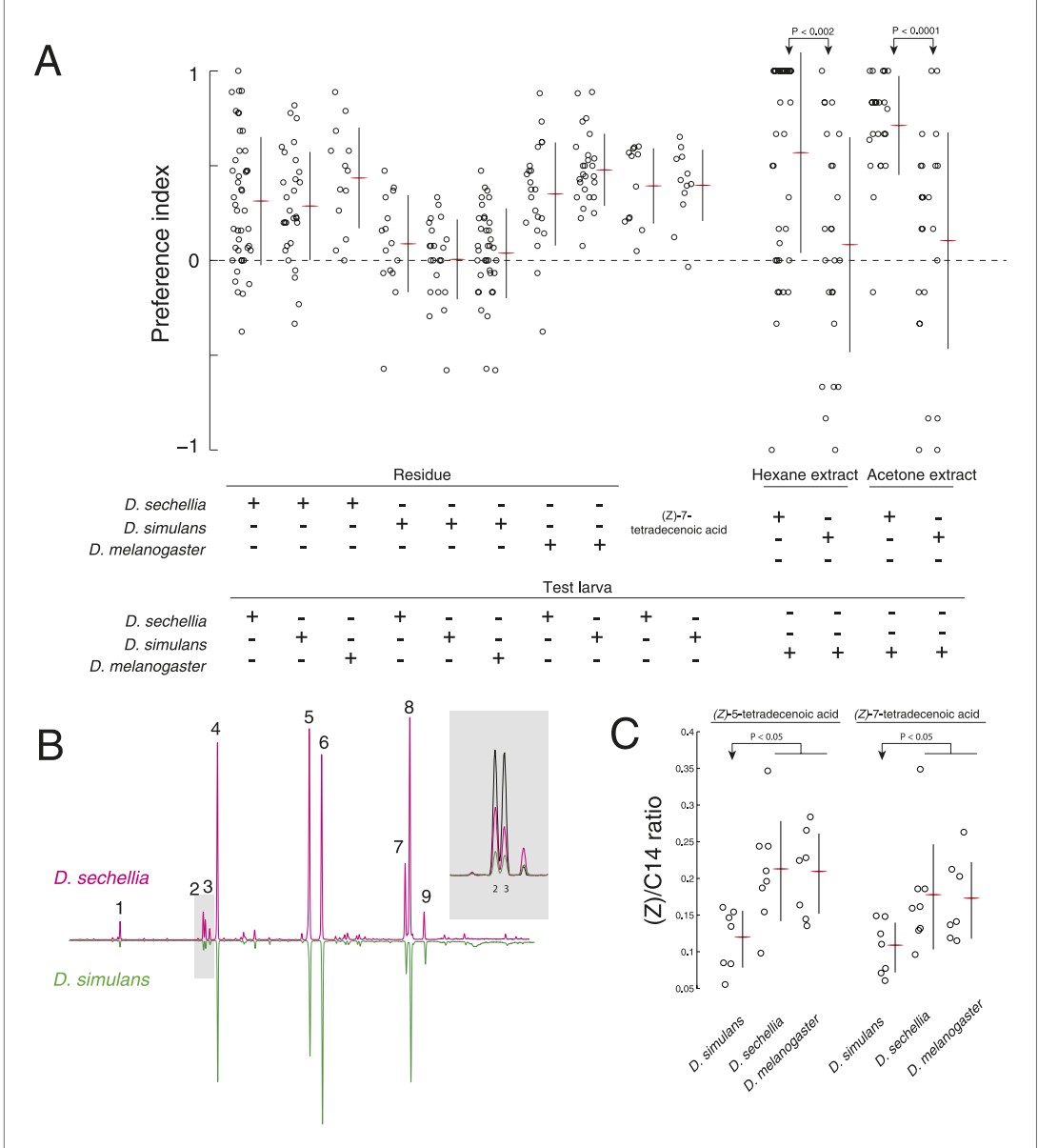

**Figure 5**. A reduction of (*Z*)-5-tetradecenoic acid and (*Z*)-7-tetradecenoic acid production underlies an evolved difference in attractive pheromone signaling. (**A**) The attraction of *D. sechellia*, *D. simulans*, or *D. melanogaster* larvae to larval residue from each species, to synthetic (*Z*)-7-tetradecenoic acid, or to extracts of larval residue. Data points for residue and extract experiments represent preference index scores of individual larvae using the assays described in *Figure 1A*, and *Figure 2A*, respectively. Datapoints measuring the response to synthetic (*Z*)-5-tetradecenoic acid and (*Z*)-7-tetradecenoic acid represent the preference of 8–12 larvae using the checkerboard assay (*Figure 3A*). Two-way ANOVA of residue experiments reveals a significant effect of residue, but not of test larvae (2-way ANOVA; residue $F_{3,199} = 40.0$, $p < 0.0001$). The residue from *D. simulans* is not attractive to any of the three species. Both *D. sechellia* and *D. simulans* are attracted to synthetic (*Z*)-7-tetradecenoic acid. *D. melanogaster* larvae are attracted to hexane and acetone extracts from *D. sechellia* larval residue, but not *D. simulans* larval residue. (**B**) Chromatographs obtained from *D. sechellia* (magenta) and *D. simulans* (green, inverted) larval hexane extracts. (1) dodecanoic acid, (2) (*Z*)-5-tetradecenoic acid, (3) (*Z*)-7-tetradecenoic acid, (4) tetradecanoic acid, (5) (*Z*)-9-hexadecanoic acid, (6) hexadecanoic acid, (7) (*Z,Z*)-9,12-octadecadienoic acid, (8) (*Z*)-9-octadecanoic acid, (9) octadecanoic acid, methyl ester. Inset, a portion of the *D. sechellia* (magenta), *D. simulans* (green), and *D.melanogaster* (black) chromatograph traces showing the relative quantities of tetradecenoic acid isomers (*Z*)-5-tetradecenoic acid (2), (*Z*)-7-tetradecenoic acid (3), and (*Z*)-9-tetradecenoic acid. (**C**) Quantification of the amount of (*Z*)-5-tetradecenoic and (*Z*)-7-tetradecenoic acid relative to saturated tetradecanoic acid in *D. simulans, D. sechellia*, and *D. melanogaster* extracts. Both (*Z*)-5-tetradecenoic acid and (*Z*)-7-tetradecenoic acid are reduced in *D. simulans* relative to *D. melanogaster* and *D. sechellia*.

*Figure 5. Continued on next page*

*Figure 5. Continued*

The following source data and figure supplement is available for figure 5:

**Source data 1**. Summary of gas chromatographs of extract from larval residue.
**Figure supplement 1**. *D. sechellia* and *D. simulans* larvae are attracted to (*Z*)-5-tetradecenoic acid.

their food resource more efficiently in aggregates (*Allee and Bowen, 1932*; *Lewontin, 1955*; *Wertheim et al., 2002*). Aggregation could also serve to reduce the chance of predation or parasitism of individuals by ants or wasps, which is a major cause of larval lethality (*Hassell, 2000*). Alternatively, larvae may release these pheromones in trails as a navigational cues for conspecifics, reinforcing the positional information provided by noisy or weak food odors.

## How are larval pheromones detected?

We found that two members of the *pickpocket* family of *DEG/ENaC* channel subunit genes, *ppk23* and *ppk29,* are required for detecting (*Z*)-5-tetradecenoic and (*Z*)-7-tetradececoic acid. These genes play broad, non-redundant roles in sex pheromone-mediated behaviors, courtship and aggression, and are required for detecting the male pheromones, cVA, (Z)-5-pentacosene, and (Z)-7-tricosene, and the female pheromones, 7,11-heptacosadiene and 7,11-nonacosadiene (*Lu et al., 2012*; *Thistle et al., 2012*; *Toda et al., 2012*; *Yuan et al., 2013*). Each of these molecules are derived from long chain fatty acids and have relatively simple double bond structures, which raises the possibility that *ppk23* and *ppk29* detect features that are common to these pheromones. How members of the *ppk* family mechanistically contribute to pheromone detection has not been resolved. Members of this family form heteromeric protein complexes (*Eskandari et al., 1999*; *Benson and et al., 2002*). It has been proposed by several research groups that the responses to specific pheromones could be conferred by other channel subunits or by an accessory protein that forms a complex with *ppk23* and/or *ppk29*. Alternatively, *ppk23* and *ppk29* may act more generally in pheromone transduction or to maintain the excitability of neurons. Recent work has demonstrated that *ppk29* is expressed more broadly than initially characterized, and that *ppk29* mRNA regulates a second channel gene (*seizure*) to control membrane excitability of motor neurons through a protein independent mechanism (*Zheng et al., 2014*). In either case, our work broadens the roles that *ppk23* and *ppk29* play in detecting pheromones. It is likely that other receptor genes are also expressed in *R58F10* and confer its specificity for larval pheromones.

## *R58F10* is an entry point to dissect pheromone processing in the SEZ

Until our work, only one *Drosophila* pheromone signaling system where both the pheromone and a single discrete class of sensory neurons that mediate a pheromone-driven behavior had been described. This is the adult male-derived sex pheromone cis-vaccenyl acetate (cVA) and the *Or65a/Or67d*-expressing olfactory receptor neurons that signal both female attractiveness and elicit male–male aggression in adult flies. cVA is a volatile pheromone, and its detection is processed via the DA1 glomerulus in the antennal lobe (*Fernández and Kravitz, 2013*). The fact that cVA-driven behaviors are elicited by a single class of sensory neurons has allowed further anatomical and functional dissection of the downstream circuit, including identification of sexually dimorphic aspects of cVA processing (*Datta et al., 2008*; *Ruta et al., 2010*; *Kohl et al., 2013*).

In contrast, there is currently a much poorer understanding of how contact-mediated pheromones are detected and decoded by downstream circuits. These pheromones, unlike cVA, are detected by gustatory neurons and the signals are processed in the subesophagael zone and ventral nerve cord (*Thorne et al., 2004*). Some of the genes and sensory neurons required to detect these pheromones have been described (*Dahanukar and Ray, 2011*) and in a few cases, the sensitivity of some pheromone-sensing neurons to contact-mediated signals has been characterized using physiology and/or calcium imaging in the periphery (*Lacaille et al., 2007*; *Thistle et al., 2012*; *Toda et al., 2012*). However, the genes and other genetic tools used to manipulate these sensory neurons are expressed broadly in multiple cell types and body regions (*Thorne et al., 2004*; *Miyamoto and Amrein, 2008*; *Moon et al., 2009*; *Weiss at al., 2011*; *Liu et al., 2012*; *Lu et al., 2012*; *Starostina et al., 2012*; *Thistle et al., 2012*; *Toda et al., 2012*; *Fan et al., 2013*), and some of these genes are required also for detecting aversive compounds other than pheromones (*Moon et al., 2009*; *Lee et al., 2010*;

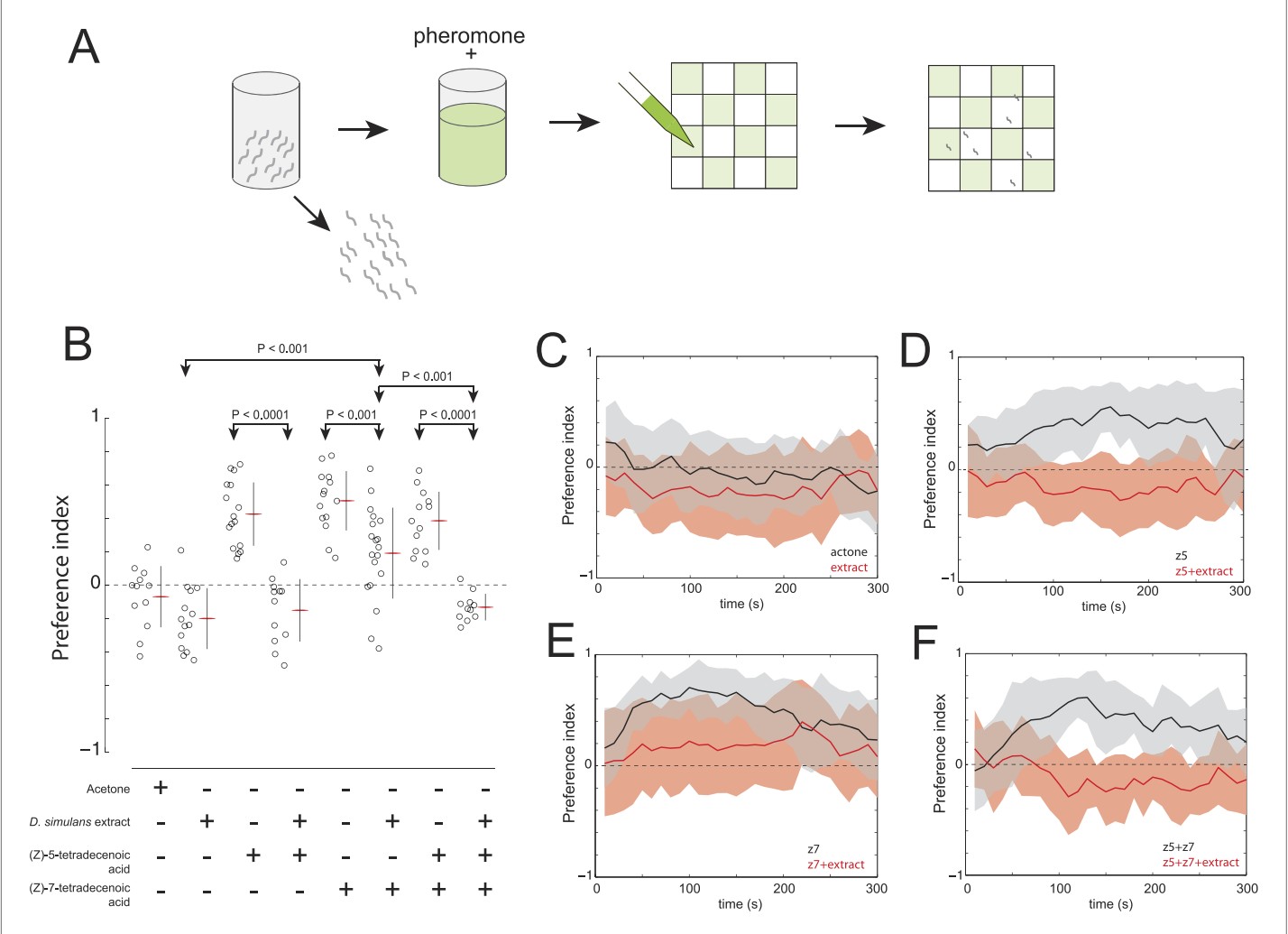

**Figure 6**. The evolved change in attractiveness of *D. simulans* is caused by the reduced production of (*Z*)-7-tetradecenoic acid and production of an unidentified repellent cue. (**A**) A schematic of an assay to measure the effect of larval pheromones in the context of *D. simulans* extract. *D. simulans* cues were extracted off glass treated with larvae and then supplemented with (*Z*)-5-tetradecenoic acid, (*Z*)-7-tetradecenoic acids or an equimolar mixture of the two compounds, and then presented to larvae using a checkerboard assay. (**B**) The attraction of *D. simulans* larvae to acetone, *D. simulans* acetone extract, *D. simulans* acetone extract supplemented with 0.5 nmol/cm² synthetic (*Z*)-5-tetradecenoic acid, 0.5 nmol/cm² synthetic (*Z*)-7-tetradecenoic acid, *D. simulans* acetone extract supplemented with synthetic (*Z*)-7-tetradecenoic acid, a mixture of synthetic (*Z*)-5-tetradecenoic acid and (*Z*)-7-tetradecenoic acid, and *D. simulans* extract supplemented with the pheromone mixture. Treatments displayed significant heterogeneity (ANOVA, $F_{7,103} = 28.1$, $p < 0.0001$). Data points are the preference index scores of small groups (10-15 individuals) of larvae in response to each condition. (**C–F**) The mean preference index score over time for the experiments summarized in (**B**). Dark lines represent the mean preference index score for each timepoint. Shaded areas represent one standard deviation. (**B**, **C**) *D. simulans* larvae are weakly repulsed by *D. simulans* residue (one-sample t-test; $p < 0.01$). (**B**, **D**) *D. simulans* extract blocks the attractive effect of synthetic (*Z*)-5-tetradecenoic acid resulting in weak repulsion (one-sample t-test; $p < 0.001$) (**B**, **E**) The supplementation of synthetic (*Z*)-7-tetradecenoic acid partially rescues the activity of *D. simulans* extract. (**B**, **F**) *D. simulans* extract blocks the activity of a mixture of (*Z*)-5-tetradecenoic and (*Z*)-7-tetradecenoic acid resulting in weak repulsion (one-sample t-test; $p < 0.001$).

*Weiss et al., 2011*; *Ling et al., 2014*). In addition, the complicated pattern of central innervation from multiple cells types in the subesophagael zone and ventral nerve cord has made it difficult to identify the subset of these cells that are pheromone responsive, and to characterize the downstream circuits.

Our study directly links the detection of two novel contact-mediated pheromones—(*Z*)-5-tetradecenoic and (*Z*)-7-tetradecenoic acid—by the receptor neuron *R58F10* to larval attraction. Furthermore, because *R58F10* labels only a single pair of bilateral sensory neurons, our work provides the first clean entry point to determine how pheromones are processed in the subesophageal zone.

## How and why has larval pheromone signaling evolved?

We found that compounds deposited by larvae of related species in the *D. melanogaster* subgroup differ in their attractiveness to other larvae. *D. simulans* has evolved in at least two ways. First, the reduced attractiveness of *D. simulans* is caused, at least in part, by a reduction in (Z)-7-tetradecenoic acid production. Second, an unidentified repellent cue present in *D. simulans* residue blocks the attractive activity of the reduced levels of (Z)-5-tetradecenoic acid, and may act through a (Z)-5-tetradecenoic acid-dependent mechanism to block larval attraction to (Z)-7-tetradecenoic acid. Interactions between pheromones that cause behaviorally antagonistic effects, such as we have observed, have been studied most intensively in the context of plume following during moth courtship (*Baker, 2008*). In many species, female moths release blends of volatile attractive pheromones to attract courting males. The major components of these blends are often shared in closely related sympatric species of moths, while these blends differ in minor components. These species-specific minor components can block the attraction of males to the blend (*Vickers and Baker, 1992*; *Quero and Baker, 1999*; *Lelito et al., 2008*), a mechanism that likely reduces interspecific courtship. An alternative hypothesis that can explain the evolved difference in larval attraction that we observed is that larvae are exquisitely sensitive to the ratio of (Z)-5-tetradecenoic acid to (Z)-7-tetradecenoic acid. We believe that this explanation is unlikely, because, although the levels of both pheromones are reduced in *D. simulans*, the ratio of these two pheromones is not significantly different from the ratios observed in *D. sechellia* and *D. melanogaster* (*Figure 6C*). These results illustrate how a relatively simple change in pheromone profiles can have a profound effect on the social behavior of a species.

It is not clear why *D. simulans* has evolved to be less attractive to other larvae. It is possible that *D. simulans* has evolved to avoid costs associated with generating aggregation cues (*Wertheim et al., 2002*, *2003*). For example, aggregation could lead to an increase in intra-specific competition over food sources, which may outweigh the density-dependent advantage larvae experience while clustered. Alternatively, the costs associated with providing positional information by releasing pheromones, such as the increased risks of predation, parasitism, or even cannibalism by conspecific larvae (*Hassell, 2000*; *Vijendravarma et al., 2013*) might in some situations outweigh the benefits of aggregation. Finally, the geographic range of *D. simulans* is largely overlapping with other species in this subgroup, and larvae of different species can often be found exploiting the same resource. While *D. simulans* produces little (Z)-5-tetradecenoic acid or (Z)-7-tetradecenoic acid, it is still attracted to these molecules. When sharing habitat with heterospecific larvae, it is possible that *D. simulans* larvae could benefit from eavesdropping on aggregation signals produced by other species while avoiding the direct and indirect costs of producing these compounds themselves.

Organisms, such as nematodes, or life stages, such as *Drosophila* larvae, that possess relatively simple behavioral repertoires may be thought to have relatively simple social lives. However, recent studies have revealed rich detail about their social environments. For example, nematodes produce a complex set of pheromone signals mediated by a diverse family of molecules called ascarisides (*Braendle, 2012*). These molecules act to coordinate aspects of development and serve as both aggregation and dispersal cues (*Macosko et al., 2009*; *Yamada et al., 2010*; *Srinivasan et al., 2012*). Both the synthesis and behavioral responses to these pheromones are evolving rapidly, and may be adaptations related to ecological specialization of these nematodes (*Braendle, 2012*; *Choe et al., 2012*). Our identification of *Drosophila* larval aggregation pheromones represents a similarly unexpected level of complexity in invertebrate behavior, and may be only the first example of multiple pheromones that mediate larval social experience in *Drosophila*.

## Materials and methods

### Fly strains and husbandry

*Drosophila* stocks were maintained under standard laboratory conditions. Stocks used were *Or83b-GAL4* (provided by T. Lee), *orco¹* (*Larsson et al., 2004*), *Gr32a-VP16* and *Gr32a¹* (*Miyamoto and Amrein, 2008*), *Gr33a-GAL4* and *Gr33a¹* (*Moon et al., 2009*), *ppk23-GAL4, Δppk23, UAS-ppk23, ppk29-GAL4, Δppk29, and UAS-ppk29* (*Thistle et al., 2012*), *attP2(UAS_unc84-2XGFP)* (provided by GL Henry), *UAS-TNT-E, UAS-IMPTV₁ₐ* (*Sweeney et al., 1995*), *yw; P{w[+mC]=UAS-mCD8::GFP.L} LL5*, and the Janelia enhancer fragment collection (*Jenett et al., 2012*). Canton S was used as wild-type *D. melanogaster*. *20XUAS-GCAMP6-S (attP40)* was obtained from the Janelia GENIE project. *D. sechellia 14021-0248.28* was obtained from the *Drosophila* Species Stock Center at

the University of California, San Diego. *D. simulans 5* is an isofemale stock collected in Princeton, New Jersey by M. Womack.

## Statistics

Data were plotted using the Matlab script errorbarjitter, available at http://www.mathworks.com/matlabcentral/fileexchange/33658-errorbarjitter. For all comparisons, we performed ANOVAs with significance values estimated using a permutation approach (*Anderson, 2011*) using custom MATLAB scripts: http://www.mathworks.com/matlabcentral/fileexchange/44307-randanova1 and http://www.mathworks.com/matlabcentral/fileexchange/44308-randanova2. P-values for comparisons of individual treatments were estimated using a Tukey honest significant difference test for multiple comparisons.

## Behavioral Assays

All behavioral responses were recorded from early third instar larvae at 25–26°C and 50% humidity. Larvae were isolated using standard procedures (*Louis et al., 2008*). All video recordings were performed with SONY DCR-HC52 MiniDV Handycam Camcorders, backlit with a white lightboard. To measure the activity of larval residue, 100 mm × 15 mm polystyrene petri dishes (Fisher Scientific International, Inc., Hampton, New Hampshire) filled with 2% agarose were treated with a high density of larvae, nearly coating the entire surface for 30 min. The larvae were removed and discarded, and half of the larval-treated agarose was cut away and replaced with untreated 2% agarose. Single test larvae were deposited in the center of each prepared assay plate and their locations were recorded for 10 min. Larval positions were analyzed afterward by hand in 10 s intervals. To isolate and assay the activity larval extract, we treated pre-washed WHEATON Glass 20 ml Scintillation Vials with isolated larvae standardized by volume for 30 min. We removed the larvae and extracted the larval residue off the glass surface for 20 min with 2 ml (for behavioral assays) or 200 µl (for chemical analysis) solvent. To measure the activity of extracts, we coated half of an agarose petri dish with extract and half with solvent. We deposited a single larva in the center of each prepared assay plate and recorded their locations for 10 min. Larva positions were determined afterward by hand in 30 s intervals. To test the attraction of larvae to individual compounds, each synthetic compound was solubilized in acetone and deposited onto a Corning 245 mm Square BioAssay Dish filled with 2% agarose. We used a 16 square stainless steel stencil adhered to the agarose to print a checkerboard pattern of control and treated squares. The dimensions of each square was 25 mm × 25 mm. Small groups (8–12) of larvae were place in the center of the assay area and monitored for 5.5 min. Larvae were tracked and preference indexes were calculated using custom MATLAB software (*Supplementary file 1*) and using the MATLAB Image Processing Toolbox. Dodecanoic acid (L4250-100G), tetradecanoic acid (M3128-10G), (*Z*)-9-hexadecenoic acid (P9417-100MG), hexadecanoic acid (P0500-10G), (*Z*)-9-octadecenoic acid (O1008-5G), and (*Z,Z*)-9,12-octadecanoic acid (L1376-1G) were obtained from Sigma-Aldrich, Inc. St. Louis, MO (*Z*)-5-tetradecenoic acid and (*Z*)-7-tetradecenoic acid were synthesized by Shanghai Medicilon, Inc., Shanghai, China.

## Histology

Larval anterior sensory organs and brains were dissected and stained using standard protocols (*Colomb et al., 2007*) with 7E8A10 anti-elav (Developmental Studies Hybridoma Bank, Iowa City, Iowa), nc82 anti-bruchpilot (Developmental Studies Hybridoma Bank), anti-GFP (Millipore, Billerica, MA), and Alexa Fluor dyes (1:200, Invitrogen). Samples were imaged on a Leica DM5500 Q Microscope using Leica Microsystems LAS AP software, and analyzed with Fiji (*Schindelin et al., 2012*).

## Chemical Analysis

Fatty acid methyl ester (FAME) preparation: A reagent solution was prepared by adding 20 µl of a 2M TMS diazomethane solution (Aldrich) to 1 ml of analytical grade dichloromethane (B&J). After vortexing, 200 µl of dry methanol was added and the solution was vortexed again. Although the reagent solution could be used for several days without any detectable degradation, it was prepared fresh for each set of samples.

 Each sample was dried with a gentle stream of N2, and 50 µl of the reagent solution was then added. After vortexing, samples were placed in an 80°C oven for 30 min. After cooling to room temperature the samples were again dried down with a gentle stream of N2 and re-dissolved in 50 µl of dichloromethane for EI-GC/MS analyses. Known amounts of fatty acid standards were prepared and analyzed in the same way. Double bond locations were established by GC/MS after conversion of the

methylesters to pyrrolidideds following the procedures of Andersson (*Andersson, 1978*). The method was modified for small sample analyses as follows: A reagent stock solution was prepared by adding 10 µl of glacial acetic acid (Sigma/Aldrich) to 1 ml of dry pyrrolidine (Sigma/Aldrich). The dichloromethane was removed from each FAME sample (including standards) and dried by a stream of N2; 50–100 µl of the pyrrolidide reagent was then added. The capped samples were heated in an 80°C oven for 1 hr and afterward allowed to cool to room temperature. Then 500 µl of dichloromethane was added and the solution was extracted 2× with 500 µl of water. The organic phase was dried by a stream of N2, and 50–100 µl of dichloromethane was then added. After vortexing, the samples were analyzed by EI-GC/MS.

Known amounts of fatty acid standards were prepared and analyzed in the same way as FAME and pyrrolidies. Tentatively indentified fatty acids were confirmed by synthesis followed by GC/MS analyses as FAME and pyrrolidide derivatives.

GC/MS FAME analyses: All samples were injected as 1-µl aliquots of dichloromethane extracts onto a gas chromatograph (HP 6890) equipped with 30 m length, 0.25-mm internal diameter, 0.25-µm film thickness DB-1 capillary column (Agilent, Palo Alto, CA, USA), interfaced to a 5973 Mass Selective Detector, operating in electron impact mode. Samples were introduced by splitless injection at 240°C. The Oven was held at 30°C for 1 min after injection and then increased 10°C/min to 260°C and held at 260°C for 6 min. The carrier gas was helium at an average velocity of 30 cm/s. An EI spectra library search was performed using the NIST11 library. When available, mass spectra and retention times were compared to those of known standards.

To better facilitate separation and thus MS identification of specific unsaturated FAMEs, the GC temperature program was changed to 30°C for 1 min after injection and then programmed for a temperature increase of 15°C/min to 150°C followed by 5°C/min to 260°C and held at 260°C for 4 min. To analyze the heavier pyrrolidide derivatives the injector temperature was increased to 260°C. The GC oven was held at 30°C for 1 min after injection and then increased 20°C/min to 200°C then by 5°C/min to 280°C and held at 280°C for 4.5 min. The EI spectra were interpreted using the NIST library and standard procedures for pyrrolidide interpretation (*Andersson and Holman, 1974*; *Andersson, 1978*).

## Activity imaging

The cuticle of third instar larvae was too thick to observe GCAMP fluorescence in DOG neurons, but signal could be observed through second instar larva cuticle. The activity of pheromone-sensing neurons was performed on second instar larvae carrying two copies of the Janelia enhancer R58F10-GAL4 and two copies of 20XUAS-GCaMP6-S (the slow, sensitive variant). Larvae were isolated and washed as described above. Hydrophobic test compounds were mobilized in adult hemolymph saline (AHS) (*Wang et al., 2003*) using detergent, CYMAL-7 (1.5 critical micelle count (CMC)) (Affymetrix; C327, Santa Clara, CA). Larvae were partially dissected in AHS and mounted on a custom-built, acrylic, 75 mm × 25 mm slide, with a 2 mm slot cut down the center for test compound delivery ending in a small reservoir. Larvae were positioned perpendicular to and facing the slot and loosely fixed into position with a 22 mm coverslip (Fisher Scientific; 12-542-B.) and the slide was flushed with AHS. GCaMP fluorescence was measured by scanning at 1 Hz on a Leica DM5500 Q Microscope using Leica Microsystems LAS AP software. Neurons were imaged for 30 s to establish baseline fluorescence, after which 200 µl AHS with 1.5 CMC CYMAL-7 (control) or AHS with 1.5 CMC CYMAL-7 and 1 nM test compound were gently injected into the central slot. Recordings continued for 150 s. Changes in fluorescence were calculated using MATLAB Image Processing Toolbox and Fiji. To calculate the appropriate concentrations of pheromone to test, we posited that the larva might detect molecules within 1 mm$^2$ in behavioral assays, and within 1 mm$^3$ around its terminal organ in calcium imaging experiments. We therefore estimate that 50 fm/cm$^2$ and 1 nM amount to $3.0 \times 10^8$ molecules and $6.0 \times 10^8$ molecules, respectively.

## Acknowledgements

We thank M Louis and A Gomez-Marin for advice with video analysis in MATLAB, T Oyama and J Truman for access to the image database of Janelia enhancer-GAL4 expression in larval brains prior to publication, and T Gonen and M Iadanza for advice on detergents. We thank S Sternson and anonymous reviewers for critical comments on the manuscript, and V Jayaraman, R Franconville, Jessica Cande, Justin Crocker, Yun Ding, Ella Preger and Troy Shirangi for helpful discussions. We also thank

the Janelia Research Campus community for facilitating this work and providing an inspirational scientific environment.

# Additional information

### Funding

| Funder | Author |
|---|---|
| Howard Hughes Medical Institute | David L Stern |

The funder had no role in study design, data collection and interpretation, or the decision to submit the work for publication.

### Author contributions

JDM, Conception and design, Acquisition of data, Analysis and interpretation of data, Drafting or revising the article; CMDM, HTA, LDL, Acquisition of data, Analysis and interpretation of data; DLS, Analysis and interpretation of data, Drafting or revising the article

# Additional files

### Supplementary file

• Supplementary file 1.

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
