## [Decision Letter]

Thank you for sending your work entitled “Evolved differences in larval social behavior mediated by novel pheromones” for consideration at *eLife*. Your article has been favorably evaluated by a Senior editor, a member of our Board of Reviewing Editors, and 3 reviewers, one of whom, Anandasankar Ray, has agreed to reveal his identity.

The reviewers discussed their comments before we reached this decision, and the Reviewing editor has assembled the following comments to help you prepare a revised submission.

The reviewers are impressed overall by the interesting data presented in this study and the technical rigor with which the study has been carried out. They also raised a number of critiques that you should address in a revised manuscript, which we believe will improve further the quality of this study.

1) It is not clear whether all ppk23 neurons or only R58F10 senses (Z) 5-tetradecenoic and (Z) 7-tetradecenoic acid pheromones. In other words, is R58F10 one neuron of the ppk23 class or is it unique as suggested in the Discussion? This could be addressed by monitoring the response of all ppk23 neurons, or all sensory neurons in the head, to (Z) 5- and (Z) 7-tetradecenoic acid.

2) Monitoring GCaMP responses to (Z) 5 and (Z) 7 in R58F10 cells in the ppk23 and ppk29 mutant would be useful to demonstrate that ppk23/29 is required for the cellular response to pheromones.

3) 1 nM concentrations are used in the GCaMP experiment in Figure 3. Much lower concentrations, 50 fmol/cm2, are used in the behavioral studies in Figure 2. It would be helpful to perform a dose-response analysis with the GCaMP studies.

4) There is no Figure 4, comparable to Figure 4, showing rescue of the ppk29 mutant.

5) Figure 1—figure supplement 3 shows a continuum of responses when a collection of driver lines are used to silence neurons. The labeling patterns of these lines are not described in detail. Do all of the lines that produce phenotypes label the neuron that is labeled by R58F10? Or does silencing of other neurons produce the same phenotype? Although no further experimentation is suggested here, depending on the resolution of the available imaging data for these lines, and the specificity of their expression patterns, it may be possible to evaluate the conclusion that a single receptor neuron is necessary to respond to the attractive pheromone.

6) What were the purities of the Z-5 and Z-7 compounds that were synthesized by Shanghai Medicilon? The Shanghai Medicilon compounds are the only ones that show robust attraction, whereas all the other compounds that were purchased from Sigma showed very little attraction. The authors should test the purity of these two compounds using the Gas Chromatogram and rule out the possibility that a contaminant was imparting attraction.

---

## [Author Response]

*1) It is not clear whether all ppk23 neurons or only R58F10 senses (Z) 5-tetradecenoic and (Z) 7-tetradecenoic acid pheromones. In other words, is R58F10 one neuron of the ppk23 class or is it unique as suggested in the Discussion? This could be addressed by monitoring the response of all ppk23 neurons, or all sensory neurons in the head, to (Z) 5- and (Z) 7-tetradecenoic acid*.

We are also curious about the other *ppk23*-expressing sensory neurons. First though, we would like to point out that we have not claimed that *R58F10* is the only sensory neuron responsive to these pheromones. Rather, in the Discussion we argue that because the single cell marked by *R58F10* is necessary for this behavior, this reagent provides a unique opportunity to dissect pheromone processing in the sub-esophageal zone of the brain. Nonetheless, we agree that testing other *ppk23* expressing neurons would be interesting. With our current calcium imaging setup, it is not possible to record from all chemosensory neurons, or from all *ppk23*-expressing neurons simultaneously. Therefore, we decided to focus on two sensory neurons most similar to *R58F10* that also innervate the dorsolateral terminal organ sensillum and that express *ppk23*. These neurons are marked by *Gr33a-GAL4*, they can be identified morphologically in our recording preparation, and their cell bodies are often adjacent in the dorsal organ ganglion (Figure 1). Also, we demonstrated that these cells are not necessary for the described behavioral response to larval pheromone (Figure 1—figure supplement 1). We cannot discriminate between these two cells, so we pooled the calcium imaging data from both.

Conservatively, we can conclude that at least one of these cells is responsive to these pheromones (Figure 3). One or both of these cells may mediate other behaviors driven by (*Z*)-5-tetradecenoic acid and/or (*Z*)-7-tetradecenoic acid.

We also slightly amended the header at line 92 that read “A single receptor neuron is necessary to respond to the attractive pheromone.” Although technically correct, we were concerned that this statement might be misconstrued to mean that we are claiming that *R58F10* is the only sensory neuron necessary for this behavior. The header has been changed to “A single receptor neuron necessary to respond to the attractive pheromone”.

*2) Monitoring GCaMP responses to (Z) 5 and (Z) 7 in R58F10 cells in the ppk23 and ppk29 mutant would be useful to demonstrate that ppk23/29 is required for the cellular response to pheromones*.

We agree that these experiments would strengthen our conclusion that ppk23 and ppk29 are required for R58F10, We agree that these experiments would strengthen our conclusion that *ppk23* and *ppk29* are required for *R58F10* cells to detect these pheromones. We have therefore included the results of these experiments in Figure 4. We find that *R58F10* cells mutant for either *ppk23* or *ppk29* do not respond to either compound.

*3) 1 nM concentrations are used in the GCaMP experiment in*
Figure 3*. Much lower concentrations, 50 fmol/cm2, are used in the behavioral studies in*
Figure 2*. It would be helpful to perform a dose-response analysis with the GCaMP studies*.

We tried to use similar amounts of pheromone in these two experiments, and we thank the reviewers for pointing out that we have not made this clear. The units we used to express the concentration of pheromone used in behavior and physiology experiments are different because in the behavior experiment the pheromone was spread across a small surface and in the GCaMP assay the pheromone was mobilized in solution. We estimate that the concentration of pheromone exposed to the larval dorsal organs is similar in the two experiments. If we posit that the larva detects molecules within 1 mm^2^ around the dorsal organ in the behavior assay and within 1 mm^3^ around the dorsal organ in the calcium imaging experiments, then the concentrations used—50 fm/cm^2^ and 1 nM—translate to 3.0x10^8^ molecules and 6.0 x10^8^ molecules, respectively. Again, we did not make this clear in the original manuscript, and we now mention this in the text: “We tested a range of pheromone concentrations estimated to encompass those encountered by larvae in the previously described behavioral assays” and include a fuller explanation in the Methods: “To calculate the appropriate concentrations of pheromone to tests, we posited that the larva might detect molecules within 1 mm^2^ in behavioral assays, and 1 mm^3^ around its terminal organ in calcium imaging experiments. With this estimate 50 fm/cm^2^ and 1 nM amount to 3.0x10^8^ molecules, and 6.0 x10^8^ molecules, respectively”. In addition, we agree that a dose-response analysis may be informative and we now show GCaMP results for three concentrations of pheromones spanning from 0.1 nM to 10 nM in Figure 3.

*4) There is no*
Figure 4*, comparable to*
Figure 4*, showing rescue of the ppk29 mutant*.

We now include the results from this experiment in Figure 4. We find that expressing *ppk29* in *R58F10* in a *ppk29* mutant background partially rescues the larval attraction to (*Z*)-7-tetradecenoic acid, but not (*Z*)-5-tetradecenoic acid.

*5)*
Figure 1—figure supplement 3
*shows a continuum of responses when a collection of driver lines are used to silence neurons. The labeling patterns of these lines are not described in detail. Do all of the lines that produce phenotypes label the neuron that is labeled by R58F10? Or does silencing of other neurons produce the same phenotype? Although no further experimentation is suggested here, depending on the resolution of the available imaging data for these lines, and the specificity of their expression patterns, it may be possible to evaluate the conclusion that a single receptor neuron is necessary to respond to the attractive pheromone*.

The larval brain images for the GAL4 lines used in this screen are now available(56). For our screen, we selected GAL4 lines from the entire collection that expressed in anterior sensory neurons. The additional GAL4 lines that failed to respond to larval residue do not all express in the pheromone-receptor neuron labeled by *R58F10*, however it is possible that the effects observed in these lines result from silencing interneurons*.* None of these other GAL4 lines drive expression as sparse as *R58F10*, and they all include a combination of sensory neurons and interneurons.

*6) What were the purities of the Z-5 and Z-7 compounds that were synthesized by Shanghai Medicilon? The Shanghai Medicilon compounds are the only ones that show robust attraction, whereas all the other compounds that were purchased from Sigma showed very little attraction. The authors should test the purity of these two compounds using the Gas Chromatogram and rule out the possibility that a contaminant was imparting attraction*.

We have now included chromatogram traces for (*Z*)-5-tetradecenoic acid and (*Z*)-7-tetradecenoic acid as Figure 2—figure supplement 2.